# Capsanthin Inhibits Atherosclerotic Plaque Formation and Vascular Inflammation in ApoE^−/−^ Mice

**DOI:** 10.3390/biomedicines10081780

**Published:** 2022-07-23

**Authors:** Sungmin Kim, Yu-Ran Lee, Eun-Ok Lee, Hao Jin, Yeon-Hee Choi, Hee-Kyoung Joo, Byeong-Hwa Jeon

**Affiliations:** 1Research Institute for Medical Sciences, College of Medicine, Chungnam National University, 266 Munhwa-ro, Jung-gu, Daejeon 35015, Korea; s13845@naver.com (S.K.); lyr0913@cnu.ac.kr (Y.-R.L.); y21c486@naver.com (E.-O.L.); jinhao0508@gmail.com (H.J.); yeonhee970@gmail.com (Y.-H.C.); 2Department of Physiology, College of Medicine, Chungnam National University, 266 Munhwa-ro, Jung-gu, Daejeon 35015, Korea

**Keywords:** capsanthin, atherosclerosis, ApoE^−/−^ mice, vascular inflammation

## Abstract

Capsanthin is a red pigment and the major carotenoid component of red paprika (*Capsicum annuum* L.). However, its role in atherosclerosis is yet to be fully elucidated. This study investigated the role of dietary capsanthin in vascular inflammation in atherosclerotic mice. We evaluated the anti-atherosclerotic effects of daily oral administration of capsanthin (0.5 mg/kg of body weight/day) in apolipoprotein E-deficient (ApoE^−/−^) mice fed a Western-type diet (WD). Capsanthin treatment inhibited vascular cell adhesion molecule 1 expression and nuclear factor-κB ser536 phosphorylation in tumor necrosis factor-α-stimulated cultured endothelial cells. Dietary capsanthin significantly inhibited the WD-induced elevation in the plasma levels of total cholesterol, low-density lipoprotein cholesterol (LDL-C), and triglyceride in mice. Interestingly, capsanthin reduced aortic plaque formation and VCAM-1 expression, which is vascular inflammation, in atherosclerotic mice. In addition, the neutrophil–lymphocyte ratio, a systemic inflammatory marker, was inhibited in capsanthin-treated mice. Furthermore, capsanthin significantly reduced the levels of proinflammatory cytokines, such as TNF-α, interleukin-6, and monocyte chemoattractant protein-1, in the plasma of atherosclerotic mice. Collectively, our data demonstrate that dietary capsanthin plays a protective role against atherosclerosis in hyperlipidemic mice. This protective effect could be attributed to the anti-inflammatory properties of capsanthin.

## 1. Introduction

Atherosclerosis is a primary cause of vascular diseases such as heart failure, myocardial infarction, and stroke [1,2]. It is a pathologic condition in which an atherosclerotic plaque, formed by the deposition of cholesterol and inflammatory cells on the lining of the artery wall, causes stenosis of the inner diameter of the blood vessel. Atherosclerosis is characterized by chronic vascular inflammation in all its phases [3,4,5]. Current evidence supports the central role of vascular inflammation regulation in treating atherosclerosis [6,7,8]. Therefore, anti-inflammatory agents may be beneficial against atherosclerosis.

Endothelial activation, which is an early process during atherosclerosis, is a crucial target for vascular inflammation regulation. Vascular endothelial cells (ECs) are the primary barrier between the blood and tissues, and play a crucial role in vascular homeostasis [9,10,11]. Endothelial activation is characterized by the expression of proinflammatory molecules such as vascular cell adhesion molecule 1 (VCAM-1), which mediates the recruitment of circulating monocytes [12,13]. Monocytes migrate into the subendothelial layer of the intima and differentiate into macrophages; this is followed by their transformation into foam cells [14]. 

In several studies, atherosclerosis in animal models resulted from accelerated plaque formation due to a diet rich in nutrients, such as cholesterol, and genetic manipulations related to cholesterol metabolism [15,16,17]. The apolipoprotein E-deficient (Apo E^−/−^) mouse model is frequently used in atherosclerosis research. ApoE^−/−^ mice that were fed a Western-type diet (WD) showed hypercholesterolemia and atherosclerotic lesion progression [16,18].

Carotenoids are the most widely distributed pigments in nature, found in a variety of fruits and vegetables [19,20,21]. Among carotenoids, capsanthin is a lipophilic red pigment responsible for the red pigmentation of paprika fruits (*Capsicum annuum* L.) [22,23,24]. Capsanthin belongs to the xanthophyll class of carotenoids, such as lutein and zeaxanthin [22]. Xanthophyllic carotenoids have been reported for their ability to improve eye functionality [25,26]. Capsanthin also shows the ability to maintain intraocular pressure within a healthy range in a rat model [27]. Among other carotenoids, capsanthin is one of the most powerful antioxidants, able to scavenge radicals because of its structural characteristics [28]. In addition to the antioxidant potential of capsanthin, previous studies have reported its antitumor, skin photoprotective, antidiabetic, and anti-obesity activities in in vitro and in vivo models [29,30,31]. We have previously reported the protective effect of capsanthin in a mouse model of nonalcoholic fatty liver disease [32]. However, the role of capsanthin in atherosclerosis remains unclear.

This study aimed to investigate the role of dietary capsanthin in vascular inflammation in atherosclerotic mice. The anti-atherosclerotic effects of the daily oral administration of capsanthin in ApoE^−/−^ mice fed a WD was evaluated. Collectively, our results demonstrate the potential role of capsanthin against chronic vascular diseases, such as atherosclerosis.

## 2. Materials and Methods

### 2.1. Cell Culture and Treatments

HUVECs, human umbilical vein endothelial cells (C2517A, Lonza, Walkersville, MD, USA), were cultured in an EGM-2 (Lonza, Walkersville, MD, USA). Capsanthin and human tumor necrosis factor-α (TNF-α) were purchased from Sigma-Aldrich (St. Louis, MO, USA). The cells were pretreated for 1 h with capsanthin at a dose range of 0.5–5 µg/mL and treated with 10 ng/mL of TNF-α for 18 h. HUVECs were used for experiments between passages five and eight. 

### 2.2. Immunoblotting

Cell and aorta tissue samples were lysed using RIPA buffer (Cell Signaling Technology, Danvers, MA, USA) and then homogenized using an ultrasonic homogenizer (Hielscher, Germany). The lysates were separated by sodium dodecyl sulfate-polyacrylamide gel electrophoresis and transferred on a polyvinylidene fluoride (PVDF) membrane. Membranes were incubated with specific primary antibodies (1:1000 anti-VCAM-1 or NF-κB p65 or p-NF-κB p65 [s536] and 1:5000 anti-β-actin). Specific antibodies against VCAM-1 (R&D Systems, Minneapolis, MN, USA), nuclear factor-κB (NF-κB) p65, p-NF-κB p65 (s536) (Cell Signaling Technology, Danvers, MA, USA), and anti-β-actin (Sigma-Aldrich, St. Louis, MO, USA) were used in this study.

### 2.3. Animal Studies

Experiments were performed using male ApoE^−/−^ mice aged 8–10 weeks. Mice were fed with a normal chow diet or a Western-type diet containing 21% fat, 34% sucrose, 19.5% casein, and 0.2% cholesterol (Envigo, Madison, MI, USA) for 12 weeks. There were four experimental groups (n = 6/each group): a normal chow-diet-fed group (ND), a Western-type diet with a vehicle (corn oil, C8267, Sigma-Aldrich, St. Louis, MO, USA)-fed group (WD), a Western-type diet with capsanthin (0.5 mg/kg of body weight/day)-fed group (WDC), and a Western-type diet with atorvastatin (20 mg/kg of body weight/day)-fed group (WDA). The mice were bred with a 12 h light/dark cycle at 24 °C for 12 weeks. All procedures of animal study were approved by the Animal Experimentation Ethics Committee of Chungnam National University (202109A-CNU-174).

### 2.4. Hematological Parameter

Blood samples (0.6 mL/each mouse) were collected from the hearts of the mice, which were deeply anesthetized by intraperitoneal injections of ketamine/xylazine. Whole blood was transferred into EDTA-coated tubes for complete blood cell count tests. Complete blood cell counts were performed using an XN-V hematology analyzer (Sysmex, Kobe, Japan). 

### 2.5. Measurement of Lipid Parameters and Plasma Cytokines

To obtain plasma, whole blood was transferred into sodium heparin tubes. Plasma samples were obtained by centrifugation at 2000× *g* for 10 min and were analyzed for blood chemistry and the measurement of cytokine levels. The total cholesterol (TC), low-density lipoprotein cholesterol (LDL-C), high-density lipoprotein cholesterol (HDL-C), and triglyceride (TG) levels were measured at the KPNT Analysis Service Center (KPNT, Cheongju, Korea). The levels of plasma cytokines, including TNF-α, interleukin (IL)-1β, IL-6, and monocyte chemoattractant protein-1 (MCP-1), were measured using an immunology multiplex assay kit at the KOMA analysis service center (KomaBiotech, Seoul, Korea). 

### 2.6. Oil Red O Staining

Aortas were harvested and fixed in 10% neutral-buffered formalin solution overnight at 20 °C. To analyze the atherosclerotic lesions in the entire aorta, the aortas were then opened longitudinally from the root to the iliac arteries and fixed en face on a rubber plate. The fixed aortas were exposed to 60% isopropanol for 3 min and incubated in oil red O solution (Sigma-Aldrich, St. Louis, MO, USA), washed with double-distilled water for 5 min, and photographed under a BA210 microscope (Motic, Vancouver, BC, Canada). The total surface area and oil red O positive area were evaluated using Image J software [33].

### 2.7. Histological Analysis

For cross-sectional aortic analysis, thoracic aortas were embedded in optimal cutting temperature (OCT) compound after fixation with 10% neutral-buffered formalin solution for 24 h. The aorta, sectioned at a thickness of 3 µm, was stained with hematoxylin and eosin (H&E) and oil red O staining. H&E and oil red O staining results were captured using a Leica S9E microscope (Solms, Hessen, Germany). These images were analyzed using Image J software to measure the plaque size in the aorta [34].

### 2.8. Immunohistochemistry

The aorta sections were incubated with anti-VCAM-1 (1:100) overnight at 4 °C. The sections were then reacted with the secondary antibody in the Dako envision detection system (Agilent Technologies, Santa Clara, CA, USA) for 30 min at room temperature. After washing in phosphate buffered saline (PBS), the sections were detected with diaminobenzidine (DAB) substrate for 3 min using the Dako envision detection system and counterstained with hematoxylin. All images were captured using a Leica S9E microscope (Solms, Germany) and analyzed using Image J software.

### 2.9. Statistics

Values are presented as means ± standard error of means (SEM). Statistical significance of the differences among groups was determined by one-way ANOVA with Dunnett’s post hoc analysis. A *p* value of less than 0.05 was determined significant.

## 3. Results

### 3.1. Capsanthin Inhibits VCAM-1 Expression and NF-κB p65 (s536) Phosphorylation in TNF-α-Stimulated HUVECs

Endothelial activation, an early process during atherosclerosis, is characterized by the expression of adhesion molecules such as the VCAM-1 [11,12]. We examined whether capsanthin regulates the expression of VCAM-1 in TNF-α-stimulated HUVECs. As shown in Figure 1A, capsanthin treatment significantly inhibited VCAM-1 expression in a dose-dependent manner. A 67% (*p* < 0.05) inhibition of VCAM-1 expression was observed with 5 μg/mL capsanthin, compared to that with TNF-α treatment alone. The expression of VCAM-1 in ECs is induced by NF-κB [35]. We examined the role of capsanthin on TNF-α-induced NF-κB p65 activation in HUVECs, and the results are shown in Figure 1B. Interestingly, TNF-α-induced NF-κB p65 (serine 536) phosphorylation was inhibited in capsanthin-treated cells. The level of phosphorylated NF-κB p65 decreased by 75% (*p* < 0.05) at 5 μg/mL capsanthin treatment. The total expression of NF-κB p65 was unaffected by capsanthin treatment. These results suggest that capsanthin inhibits TNF-α-induced endothelial activation.

### 3.2. Dietary Capsanthin Regulates Plasma Lipid Levels of WD-fed ApoE^−/−^ Mice

The animal experiments were designed to investigate whether capsanthin administration affected aortic inflammation in hyperlipidemic ApoE^−/−^mice. To examine the role of capsanthin, capsanthin (0.5 mg/kg of body weight/day) was orally administered to ApoE^−/−^ mice daily. The capsanthin dose used corresponded to the dose that inhibits nonalcoholic fatty liver disease in mice [32]. The lipid-lowering agent, atorvastatin (20 mg/kg of body weight/day), was orally administered to ApoE^−/−^ mice daily. First, we evaluated whether capsanthin affects plasma lipid levels, such as TC, LDL-C, HDL-C, and TG. As shown in Table 1, ApoE^−/−^ mice in the WD group showed a remarkable increase in plasma lipid levels compared to those in the ND group. Surprisingly, the capsanthin-treated WDC group showed a significant inhibition of TC (48%, *p* < 0.001), LDL-C (55%, *p* < 0.001), and TG (36%, *p* < 0.05) in the plasma levels, compared to the WD group. 

### 3.3. Capsanthin Reduces WD-Induced Atherosclerotic Plaque Formation

We examined whether capsanthin regulated plaque formation via histological examination of the mouse aorta using oil red O or H&E staining. The WD group showed a remarkable increase in oil red O-positive lesions in whole and cross-sectional aorta, compared to the ND group (Figure 2). The relative percentage of oil red O-positive area in the total area of the ND group was 5%, whereas that of the WD group was 30% (*p* < 0.001). However, the atorvastatin-treated WDA group showed a remarkable inhibition of the plaque area compared to the WD group. Interestingly, the relative percentage of the plaque area of the capsanthin-treated WDC group was significantly reduced by 12% (*p* < 0.01), and the reducing effect of the WDC group was similar to that of the WDA group. These data indicate that dietary capsanthin has the potential to exert anti-atherosclerotic effects.

### 3.4. Capsanthin Inhibits Vascular Inflammation in Atherosclerotic Mice

Next, we examined the expression of aortic VCAM-1, a vascular inflammatory marker [36], using immunohistochemical staining. As shown in Figure 3A, VCAM-1 level was remarkably increased six-fold (*p* < 0.001) in the aorta of the WD group compared to the aorta of the ND group. VCAM-1 was expressed in the blood vessel intima and media of atherosclerotic lesions. Interestingly, the VCAM-1 level was significantly reduced by 85% (*p* < 0.001) in the aortas of the WDC group, compared to the WD group. The WDA group also showed a significant inhibition of 92%. Immunoblot analysis was performed to quantitatively analyze the alterations in VCAM-1 expression in the aorta. As shown in Figure 3B, the WD group showed a marked elevation in aortic VCAM-1 expression compared to the ND group. Upregulated aortic VCAM-1 levels in the WD group were decreased by 53% (*p* < 0.01) in the WDC group. These data indicate that capsanthin administration reduces atherosclerotic inflammation, resulting in the inhibition of plaque formation.

### 3.5. Capsanthin Attenuates the Neutrophil–Lymphocyte Ratio (NLR), an Inflammatory Marker, in Atherosclerotic Mice

We examined whether capsanthin regulated hematologic parameters using a complete blood count. The red blood cell (RBC) counts remained unchanged under the experimental conditions. (Appendix A). However, the differential WBC counts were significantly altered. As shown in Figure 4, WBC counts did not differ between the experimental groups. However, the percentage of neutrophils in WD group was upregulated (67% for WD vs. 36% for ND, *p* < 0.05), while the lymphocyte percentage was downregulated (29% for WD vs. 63% for ND, *p* < 0.05). Interestingly, the WDC group showed significant changes in the percentage of neutrophils (46%, *p* < 0.05) and lymphocytes (51%, *p* < 0.05). NLR is an inflammatory marker related to atherosclerosis [37,38]. The WD group showed an increased NLR, which was suppressed in the WDC group. The reducing effect of capsanthin was similar to that of atorvastatin. These results suggest that capsanthin exerts anti-inflammatory effects.

### 3.6. Capsanthin Inhibits Plasma Inflammatory Cytokines

Inflammatory cytokines are closely related to the progression of vascular lesions and the pathogenesis of atherosclerosis [39]. Finally, we examined whether capsanthin affects the production of circulating inflammatory factors in the blood. The levels of proinflammatory and chemotactic cytokines in mice plasma were measured. The WD group showed a dramatic increase in plasma cytokine and chemokine levels (Figure 5). Interestingly, WD-induced proinflammatory cytokines, including TNF-α, interleukin (IL)-1β, IL-6, and monocyte chemoattractant protein-1 (MCP-1), were significantly inhibited in capsanthin-treated mice. Collectively, our data suggest that capsanthin attenuates the production of circulating inflammatory factors. 

## 4. Discussion

In the present study, we investigated the role of dietary capsanthin in atherosclerotic mice. The capsanthin-mediated anti-inflammatory effects were not restricted to cultured ECs. Daily administration of capsanthin inhibited vascular inflammation and plaque formation in the arteries of ApoE^−/−^mice fed with a WD.

Carotenoids have various biological roles that contribute to their preventive and therapeutic effects, including anticancer, immunomodulatory, and anti-inflammatory effects [40,41,42]. Capsanthin is a lipophilic red pigment responsible for the red pigmentation of paprika fruits (*Capsicum annuum* L.) and belongs to the xanthophyll class of carotenoids [22,23,24]. Among other carotenoids, capsanthin is one of the most powerful antioxidants, able to scavenge radicals due to its structural characteristics [28]. However, capsanthin has not been studied as extensively as other carotenoids, such as beta-carotene, lutein, lycopene, astaxanthin, and zeaxanthin. Since the health benefits of capsanthin were only discovered recently, it is necessary to investigate its effects on various diseases.

In the present study, we orally administered capsanthin to ApoE^−/−^ mice. Reports have shown that capsanthin is not detected in human plasma under basal conditions; however, the ingestion of capsanthin-rich foods is followed by an increase in the plasma concentrations of capsanthin [43,44]. These data suggest that capsanthin is absorbed and becomes biologically available in human plasma.

This study demonstrated the inhibitory role of capsanthin on TNF-α-induced endothelial activation. Vascular ECs provide an essential protective barrier [10]. Endothelial activation is characterized by the expression of adhesion molecules that mediate the recruitment of circulating monocytes [12,13]. Our data show that capsanthin treatment inhibited the VCAM-1 expression. The mechanism of the inhibitory role of capsanthin on adhesion molecules is linked to NF-kB regulation. This study supports the finding that NF-kB p65 activation is inhibited by treating TNF-α-stimulated HUVECs with capsanthin. Endothelial activation, an early event in atherosclerosis, is an important target in regulating atherosclerotic lesions [9,10,11]. In parallel with the reduction in VCAM-1 in activated endothelial cells, a capsanthin-mediated inhibitory effect on vascular plaque formation in atherosclerotic mice was also shown.

The capsanthin-mediated alleviation of atherosclerotic lesions is hypothesized to be mainly mediated by its inhibitory effect on vascular inflammation. Atherosclerosis is a chronic vascular disease, and inflammation is involved in all of its stages. Recent studies have shown that controlling vascular inflammation plays a central role in treating atherosclerosis [6,7,8]. We have also previously reported that anthocyanin-rich extracts inhibit aortic plaque formation by inhibiting vascular inflammation in atherosclerotic mice [45]. In this study, dietary capsanthin treatment reduced vascular inflammation. In addition to inhibiting aortic VCAM-1 expression, NLR, an inflammatory marker associated with atherosclerosis, was also reduced by capsanthin treatment. Inflammatory cytokines are implicated in the progression of vascular lesions [39]. Our data revealed that capsanthin reduced the production of inflammatory cytokines in the blood. These results suggest that capsanthin exerts an anti-inflammatory effect, which plays an important role in preventing atherosclerosis progression. Based on this study, further research is needed to investigate the role of capsanthin in various diseases related to inflammation.

The capsanthin-mediated inhibition of atherosclerosis may also be associated with the lipid-lowering effects of capsanthin. Our data reveal that capsanthin reduced plasma lipid levels, including TC, LDL-C, and TG. Interestingly, there is evidence that lipid-lowering agents such as statins have anti-inflammatory effects [46,47]. This suggests that the capsanthin-mediated lipid-lowering effect may also affect the regulation of vascular inflammation, thereby inhibiting the progression of atherosclerosis. 

Our data reveal that the administration of 0.5 mg/kg of capsanthin inhibited vascular inflammation and atherosclerotic lesions in WD-fed ApoE^−/−^ mice. Based on body surface area conversion, 0.5 mg/kg of capsanthin in mice equals 0.041 mg/kg of capsanthin in humans [48]. In a previous study, the capsanthin content in Raon red paprika was 34.33 mg/100 g dry weight [32]. For a similar anti-arteriosclerosis effect, capsanthin content was converted to 72 g of red paprika per day for a person weighing 60 kg. Considering the amount of Raon red paprika, this dosage is a reasonable amount that a daily diet may provide. In addition, in terms of application, capsanthin-rich products are widely available in the form of food and feed additives as well as supplements. Based on these values, it is possible to suggest the required amount of food containing capsanthin that could effectively prevent atherosclerosis.

## 5. Conclusions

Taken together, our findings reveal that the administration of capsanthin alleviates inflammatory markers in TNF-α-stimulated cultured ECs, and prevents the progression of atherosclerosis in WD-fed ApoE^−/−^mice. This protective effect could be attributed to the anti-inflammatory properties of dietary capsanthin. Therefore, regular consumption of capsanthin-rich foods may contribute to lowering the risk of chronic vascular inflammatory diseases such as atherosclerosis. Our preclinical efficacy evaluation of capsanthin for arteriosclerosis can be a reference for future clinical studies.

## Figures and Tables

**Figure 1 biomedicines-10-01780-f001:**
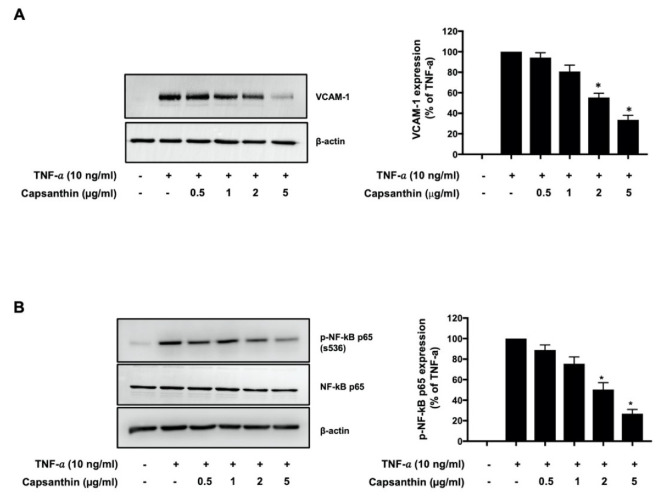
Capsanthin inhibited tumor necrosis factor−α (TNF−α) −induced vascular cell adhesion molecule 1 (VCAM−1) expression and nuclear factor−κB (NF−κB) p65 phosphorylation in human umbilical vein endothelial cells (HUVECs). (**A**) Immunoblot analysis for VCAM−1 expression using cell lysate from TNF−α−treated HUVEC for 12 h following pre-treatment with capsanthin (left). β−actin was used as a loading control for VCAM−1. The bar graphs represent the quantitative difference in the expression of VCAM−1 (right). (**B**) Immunoblot analysis for NF−κB p65 phosphorylation (ser536) using cell lysate from TNF−α−stimulated HUVEC for 6 h following pre-treatment with capsanthin (left). NF−κB p65 and β−actin were used as loading controls for NF−κB p65 phosphorylation. Bar graphs represent the quantitative difference in the expression of NF−κB p65 phosphorylation (right). All values represent the mean ± standard error of the mean (SEM) (*n* = 3). * *p* < 0.05 vs. TNF-α-induced group.

**Figure 2 biomedicines-10-01780-f002:**
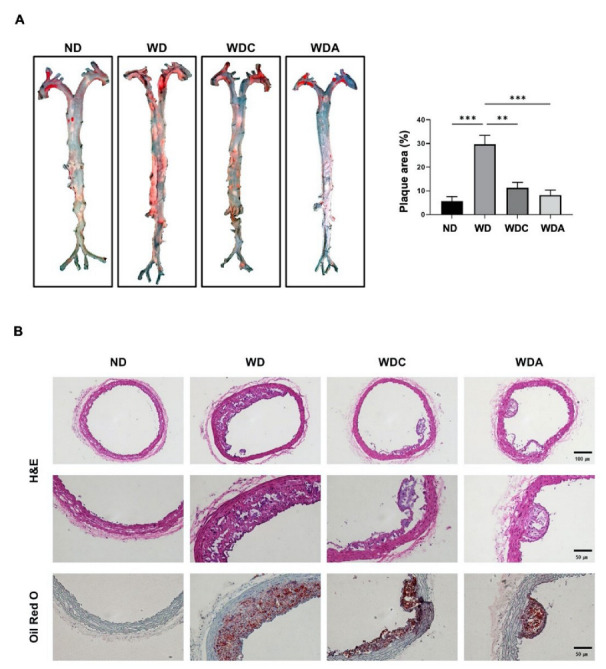
Capsanthin inhibited plaque formation in the aorta of Western-type diet (WD)-fed apolipoprotein E deficient (ApoE^−/−^) mice. (**A**) Whole aortas were stained with oil red O for the observation of lipid accumulation (red). Representative images of the aorta from each group (left). Quantification of the plaque area as a relative percentage of the oil red O-positive area in total area using image J software (right). All values represent the mean ± SEM (*n* = 3). ** *p* < 0.01, *** *p* < 0.001 vs. WD group. (**B**) Representative photomicrographs for cross-sectional aortas of H&E or oil red O-staining from the thoracic aortic from each group in ApoE^−/−^ mice. H&E images indicate ×10 magnification (upper, scale bar, 100 μm) and ×20 magnification (bottom, scale bar, 50 μm). Cross-sectional aortas were stained with Oil Red O and counterstained with hematoxylin. Oil red O-stained aorta images show ×20 magnification (scale bar, 50 μm). ND; normal diet, WD; Western-type diet, WDC; capsanthin with Western-type diet, WDA; atorvastatin with Western-type diet.

**Figure 3 biomedicines-10-01780-f003:**
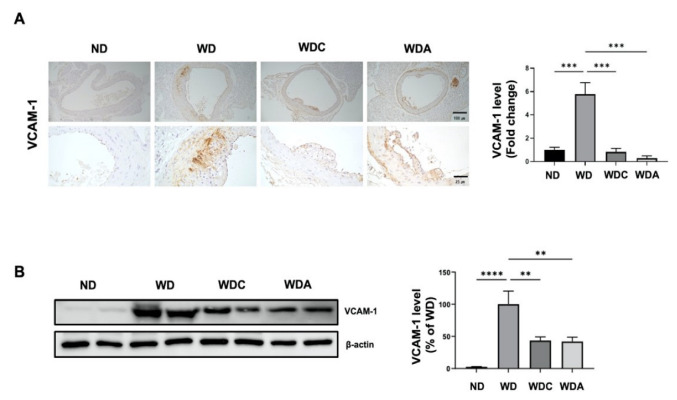
Capsanthin inhibited vascular inflammation in the aorta of Western-type diet (WD)-fed ApoE^−/−^ mice. (**A**) Photography of immunohistochemistry for VCAM-1 expression in the aortas of each group. VCAM-1 was developed with a brown color in the aortas of each group and counterstained with purple color (left). Representative immunohistochemistry images indicate ×10 magnification (upper, scale bar, 100 μm) and ×40 magnification (bottom, scale bar, 25 μm). Fold changes in the levels of VCAM-1 expression relative to those of the WD group are shown for each group (right). All values represent the mean ± SEM (*n* = 3). ** *p* < 0.01, *** *p* < 0.001 vs. WD group. (**B**) Immunoblotting for VCAM-1 expression using aortic-tissue lysates obtained from each experimental group (left). Relative band intensities were normalized to that of β-actin. Bar graphs represent the quantitative difference in VCAM-1 expression (right). All values represent the means ± SEM (*n* = 4). ** *p* < 0.01, *** *p* < 0.001, **** *p* < 0.0001 vs. WD group. ND: normal diet, WD: Western-type diet, WDC: capsanthin with Western-type diet, WDA; atorvastatin with Western-type diet.

**Figure 4 biomedicines-10-01780-f004:**
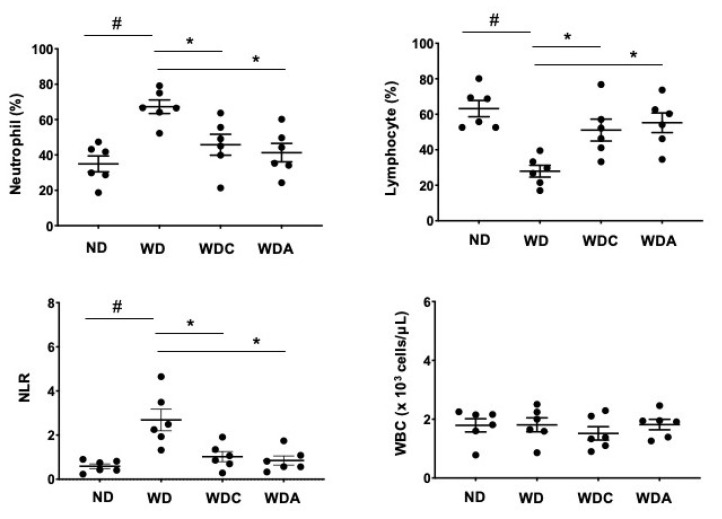
Capsanthin attenuated the neutrophil-to-lymphocyte ratio (NLR) of Western-type diet (WD)-fed ApoE^−/−^ mice. White blood cells (WBC), neutrophils, and lymphocytes were analyzed from complete blood cells using a hematology analyzer. NLR was calculated by dividing neutrophils by lymphocytes in each experimental group. All values represent the mean ± SEM (*n* = 6). # *p* < 0.05 vs. the ND group; * *p* < 0.05 vs. the WD group. ND: normal diet, WD: Western-type diet, WDC: capsanthin with Western-type diet, WDA: atorvastatin with Western-type diet.

**Figure 5 biomedicines-10-01780-f005:**
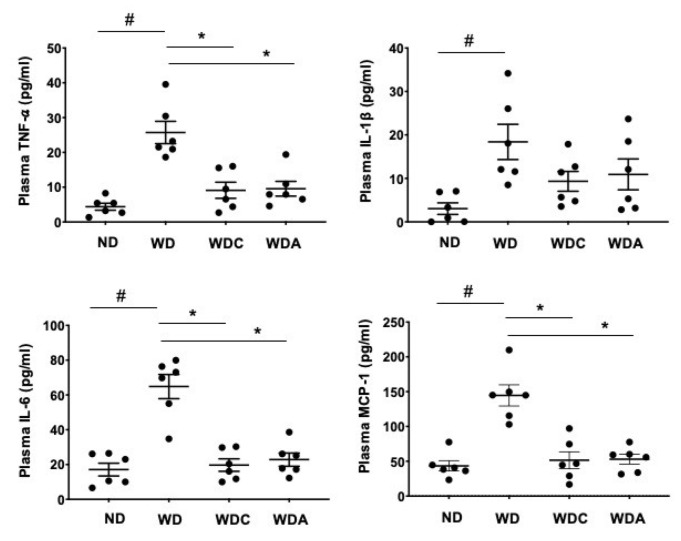
Capsanthin inhibited Western-type diet (WD)-induced proinflammatory cytokines in the plasma of ApoE^−/−^ mice. Each graph shows the plasma levels of tumor necrosis factor-α (TNF-α), interleukin-1β (IL-1β), interleukin-6 (IL-6), and monocyte chemoattractant protein-1 (MCP-1) in the experimental groups. All values represent the mean ± SEM (*n* = 6). # *p* < 0.05 vs. the ND group; * *p* < 0.05 vs. the WD group. ND: normal diet, WD: Western-type diet, WDC: capsanthin with Western-type diet, WDA: atorvastatin with Western-type diet.

**Table 1 biomedicines-10-01780-t001:** Plasma lipid levels in ApoE^−/−^ mice with normal and Western-type diets.

Plasma Lipids	ND	WD	WDC	WDA
TC (mg/dL)	447.5 ± 31.6	1470.0 ± 77.2 ^##^	764.8 ± 119.7 ***	958.3 ± 53.7 ***
LDL-C (mg/dL)	187.3 ± 20.3	823.3 ± 47.0 ^##^	374.6 ± 60.8 ***	540.9 ± 31.4 ***
HDL-C (mg/dL)	62.9 ± 7.7	99.3 ± 10.2 ^##^	63.9 ± 3.8 **	80.9 ± 3.3
TG (mg/dL)	97.0 ± 15.2	184.5 ± 24.7 ^##^	117.5 ± 14.1 *	116.0 ± 12.5 *

ND: normal diet, WD: Western-type diet, WDC: capsanthin with a Western-type diet, WDA: atorvastatin with a Western-type diet, TC: total cholesterol, LDL-C: low-density lipoprotein cholesterol, HDL-C: high-density lipoprotein cholesterol, TG: triglyceride. All values represent mean ± SEM, *n* = 6 animals per group. ## *p* < 0.01 versus ND, * *p* < 0.05, ** *p* < 0.01, *** *p* < 0.001 versus WD.

## Data Availability

Data is contained within the article and Appendix A.

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
