# Peer review of "Capsanthin Inhibits Atherosclerotic Plaque Formation and Vascular Inflammation in ApoE^−/−^ Mice"

_biomedicines, 2022, doi:10.3390/biomedicines10081780_

Round 1

Reviewer 1 Report

In this interesting study, the authors demonstrated the anti-atherosclerotic effects of capsanthin using an in vitro an in vivo experimental approach. Overall this is valuable study, in which the authors investigated the role of dietary capsanthin in vascular inflammation in atherosclerotic mice - apolipoprotein E-deficient (ApoE-/-) mice fed a Western-type diet (WD). In addition, the authors demonstrated that capsanthin can inhibit in vitro the expression of VCAM1 and NFkB Ser536 activity in activated endothelial cells. The authors demonstrated that dietary capsanthin in WD-fed mice: (i) reduced the total cholesterol, LDL-C, and triglyceride levels in plasma; (ii) reduced aortic plaque formation and VCAM-1 expression; (iii) neutrophil-lymphocyte ratio and (iv) reduced the plasma levels of TNF-a, IL-6, and MCP-1.

The authors suggested that these data provide novel insights into the potential anti-atherosclerotic role of capsanthin. The subject of the article is noteworthy, adding new data in the field of identifying new potential therapies to help treating atherosclerosis. The study has an appropriate experimental design, data presentation and discussion, has a good statistical analysis to support the conclusions.

Specific comments:

1.     Methods: how long TNF-alpha was incubated with the cells? (see line 77).

2.     Methods: what components of WD were administered to mice groups (see lines 91-93)? What are the administered doses for capsanthin and atorvastatin in the mice groups WDC and, respectively, WDA? What is the used ”vehicle” for mice treatments (see lines 92-93)?

3.     Methods: How much blood was collected from each mice (lines 98-99)?

4.     Methods: lipid parameters and detailed cytokines (as shown in results, measured with the immunology multiplex assay kit) should be presented in a new section, separated by the section 2.4. How HDL-cholesterol was measured (since it was omitted in line 104)?

5.     Results: I would suggest presenting the expression changes (fold-change or percent-change) of the measured parameters compared to reference groups presented in the manuscript, i.e. rather than simply repeating all the data included in the Table 1 (lines 171-172).

6.     All the data presented in the manuscript’s text (fold-change or percent-change) should be shown together with their p-values.

7.     Overall in the manuscript, LDL-cholesterol and HDL-cholesterol should be abbreviated accordingly, LDL-C and HDL-C. Some positions to be corrected: abstract (lines 19-20), methods (line 104), Table 1 (content and footer) and lines 169-172.

8.    Results: sections 3.3 (lines 183-189) and 3.4 (lines 201-210) need more discussion about the data presented in Figures 2 and 3a, i.e. the changes observed in the analyzed atherosclerotic plaque areas with their p-values.

9.     There are only few comments about WDA-treated mice group data. This is a positive control for the WDC treatment group but it was less presented in the manuscript’s text.

10.  Some minor English grammar and typo error occurred, i.e. in line 110 “to analysis for atherosclerotic lesion of entire aorta” it might be “to analyze the atherosclerotic lesions in entire aorta”. A deep check for these errors should be performed.

Author Response

We thank reviewer for the critical review and for the valuable comments. We corrected/implemented all the points raised, accordingly. Corrections in the revised manuscript are highlighted in red. 

Specific comments:

1. Methods: how long TNF-alpha was incubated with the cells? (see line 77).

 ->Thanks for your question. We added the incubation time to line 77 of the revised text.

2. Methods: what components of WD were administered to mice groups (see lines 91-93)?

->Thanks for your question. We added the information for components of WD to line 91-92.

What are the administered doses for capsanthin and atorvastatin in the mice groups WDC and, respectively, WDA?

->We added the information for doses of capsanthin and atorvastatin to line 95-96.

What is the used ”vehicle” for mice treatments (see lines 92-93)?

->We added the information for vehicle to line 94.

3. Methods: How much blood was collected from each mice (lines 98-99)?

->Thanks for your question. We added the information about the amount of blood collected form each mouse to line 101.

4. Methods: lipid parameters and detailed cytokines (as shown in results, measured with the immunology multiplex assay kit) should be presented in a new section, separated by the section 2.4. How HDL-cholesterol was measured (since it was omitted in line 104)?

->As you recommended, lipid parameters and cytokines were presented in a new section 2.5 (line 105-114). HDL-cholesterol was added in line 109.

5. Results: I would suggest presenting the expression changes (fold-change or percent-change) of the measured parameters compared to reference groups presented in the manuscript, i.e. rather than simply repeating all the data included in the Table 1 (lines 171-172).

->Thanks for your valuable comment. As you recommended, we presented the changes of the measured parameters as percent-change compared to WD group (line 178-180).  

6. All the data presented in the manuscript’s text (fold-change or percent-change) should be shown together with their p-values.

->Thank you for nice comment. We presented data for fold-change or percent-change along with p-value in the full text.

7. Overall in the manuscript, LDL-cholesterol and HDL-cholesterol should be abbreviated accordingly, LDL-C and HDL-C. Some positions to be corrected: abstract (lines 19-20), methods (line 104), Table 1 (content and footer) and lines 169-172.

->As you recommended, incorrected sentences were modified to LDL-C or HDL-C in the following parts: abstract (line 20), methods (line 109), table 1 (line 182-186), line 176-179, and 322.

8. Results: sections 3.3 (lines 183-189) and 3.4 (lines 201-210) need more discussion about the data presented in Figures 2 and 3a, i.e. the changes observed in the analyzed atherosclerotic plaque areas with their p-values.

->As you recommended, data was presented as percent-change or fold-change along with p-value. Additionally, discussion of data has been added to sections 3.3 (line 193-198) and 3.4 (line 214-219).

9. There are only few comments about WDA-treated mice group data. This is a positive control for the WDC treatment group but it was less presented in the manuscript’s text.

->Thank you for nice comment. Atorvastatin is usually used as lipid-lowering agent. In this study, WDA group showed inhibitory effect of plaque formation and vascular inflammation. As you recommended, we added the explain for WDA group data to line 174-175, 194-195, and 218-219.  

10. Some minor English grammar and typo error occurred, i.e. in line 110 “to analysis for atherosclerotic lesion of entire aorta” it might be “to analyze the atherosclerotic lesions in entire aorta”. A deep check for these errors should be performed.

->Thank you. As you recommended, we revised the sentence to “to analyze the atherosclerotic lesions in entire aorta” in line 117. Throughout the manuscript, typographical error was re-checked and corrected.

Reviewer 2 Report

The author revealed that the administration of capsanthin alleviates in flammatory markers in TNF-stimulated cultured ECs and prevents atherosclerosis progression in WD-fed ApoE-/-mice. This protective effect could be attributed to the anti-inflammatory properties of dietary capsanthin. Therefore, regular consumption of capsanthin-rich foods may contribute to lowering the risk of chronic vascular inflammatory diseases such as atherosclerosis. Albeit, the author provides an insight into preclinical efficacy evaluation of capsanthin for arteriosclerosis, I still have some suggestions.

1, All figures are highly professional, and the authors should guide the readers to the meaning of the images appropriately; otherwise, it is likely to cause misunderstandings. Therefore, I suggest that the author consider revising these figure legends again.

2, The scale bar was quite vague in Fig2B and 3C, please also upload the high resolution figure during revision.

3, There are a few typo issues for the authors to pay attention. Please unify the writing of scientific terms. “Italic, capital” ? make it consistent throughout the whole manuscript, and the manuscript also needs English proofreading.

Author Response

We thanks for the critical review and for the valuable comments. We corrected/implemented all the points raised, accordingly. Corrections in the revised manuscript are highlighted in red. 

1, All figures are highly professional, and the authors should guide the readers to the meaning of the images appropriately; otherwise, it is likely to cause misunderstandings. Therefore, I suggest that the author consider revising these figure legends again.

->Thanks for nice suggestion. As you recommended, figure legends were modified to better represent the meaning of the image.  

2, The scale bar was quite vague in Fig2B and 3C, please also upload the high resolution figure during revision.

->Thanks for nice comment. We changed the figure 2 and figure 3 with high resolution images.

3, There are a few typo issues for the authors to pay attention. Please unify the writing of scientific terms. “Italic, capital” ? make it consistent throughout the whole manuscript, and the manuscript also needs English proofreading.

->Thank you for valuable comment. Throughout the manuscript, typographical error was re-checked and corrected. Scientific terminology was also reviewed.